# The Low-Temperature Photocurrent Spectrum of Monolayer MoSe_2_: Excitonic Features and Gate Voltage Dependence

**DOI:** 10.3390/nano12030322

**Published:** 2022-01-19

**Authors:** Daniel Vaquero, Juan Salvador-Sánchez, Vito Clericò, Enrique Diez, Jorge Quereda

**Affiliations:** Nanotechnology Group, USAL—Nanolab, Universidad de Salamanca, E-37008 Salamanca, Spain; danivaqu@usal.es (D.V.); juan2s@sal.es (J.S.-S.); vito_clerico@usal.es (V.C.); enrisa@usal.es (E.D.)

**Keywords:** excitons, transition metal dichalcogenides, photocurrent spectroscopy

## Abstract

Two-dimensional transition metal dichalcogenides (2D-TMDs) are among the most promising materials for exploring and exploiting exciton transitions. Excitons in 2D-TMDs present remarkably long lifetimes, even at room temperature. The spectral response of exciton transitions in 2D-TMDs has been thoroughly characterized over the past decade by means of photoluminescence spectroscopy, transmittance spectroscopy, and related techniques; however, the spectral dependence of their electronic response is still not fully characterized. In this work, we investigate the electronic response of exciton transitions in monolayer MoSe_2_ via low-temperature photocurrent spectroscopy. We identify the spectral features associated with the main exciton and trion transitions, with spectral bandwidths down to 15 meV. We also investigate the effect of the Fermi level on the position and intensity of excitonic spectral features, observing a very strong modulation of the photocurrent, which even undergoes a change in sign when the Fermi level crosses the charge neutrality point. Our results demonstrate the unexploited potential of low-temperature photocurrent spectroscopy for studying excitons in low-dimensional materials, and provide new insight into excitonic transitions in 1L-MoSe_2_.

## 1. Introduction

Two-dimensional transition metal dichalcogenides (2D-TMDs) are an ideal material platform for exciton physics. This family of materials presents unusually large exciton binding energies and lifetimes, even at room temperature; consequently, their optical and optoelectronic properties are largely dominated by excitonic transitions [1]. Indeed, the discovery of this family of materials has brought renewed hopes to the development of excitonic devices capable of operating at room temperature.

Excitonic devices based on 2D-TMDs often rely on the so-called bright exciton states, i.e., exciton states that are capable of emitting light upon relaxation. These exciton states are typically studied and characterized via photoluminescence spectroscopy and related techniques. However, in order to fully understand exciton dynamics in 2D-TMDs, it is also necessary to have access to non-radiating excitonic states. In consequence, in recent years, different characterization techniques—such as absorption spectroscopy [2,3,4,5,6] or electroluminescence spectroscopy [7]—have increasingly gained popularity due to their potential for investigating excitonic states not accessible via PL.

As we demonstrated in a recent publication [8], low-temperature photocurrent spectroscopy (PCS) [9,10,11] provides another simple, powerful, and yet largely underused complementary approach for studying excitonic transitions in 2D-TMDs. The introduction of the low temperature in this technique helps to reduce thermal disorder, obtaining resolution bandwidths of around 10 meV, comparable with low-temperature PL spectroscopy. PCS not only enables the observation of excitonic transitions with linewidths comparable to those obtained in PL measurements, but also provides the possibility of detecting exciton transitions that cannot be easily observed via PL due to the presence of dominant non-radiative relaxation mechanisms. Furthermore, since excitons are charge-neutral, their contribution to photocurrent requires them to dissociate without electron–hole recombination. Consequently, PCS could enable the extraction of information on the dynamics of non-radiative exciton dissociation mechanisms. 

PCS characterization is particularly relevant for excitonic devices aiming to interface light-based and electron-based communication protocols, as it provides a route towards the electrical detection of optically excited exciton states. However, PCS measurements in some of the most technologically relevant 2D-TMDs are still scarce in the scientific literature.

In this work, we report on low-temperature PCS measurements in high-quality monolayer (1L) MoSe_2_ phototransistors. Our measurements allow us to identify the spectral features associated with the main exciton and trion transitions of monolayer MoSe_2_, with spectral bandwidths as low as 15 meV. We also characterize the evolution of the photoresponse as a function of the Fermi level, observing a very strong modulation of the photocurrent, which even switches from positive to negative as the Fermi level approaches the edge of the valence band. The results presented here provide new insight into excitonic transitions in 1L-MoSe_2_, bearing great importance for the development of excitonic devices based on this material.

## 2. Materials and Methods

### 2.1. Device Fabrication

Figure 1 summarizes the main steps of the monolayer MoSe_2_ phototransistor fabrication, following a similar procedure to the one described by Quoc An Vu et al. [12]. We started the process with the exfoliation of the MoSe_2_ by micromechanical cleavage. We used high-quality tape (BT-150E-CM, Nitto) to exfoliate the bulk crystals, and we placed the flakes obtained in a polydimethylsiloxane (PDMS) substrate. Then, we inspected the substrates through optical microscope, identify the 1L-MoSe_2_ flakes by their optical contrast and confirm their thickness by micro-Raman spectroscopy [13] as further detailed in Appendix A. The thickness of 1L-MoSe_2_ was found to be around 0.7 nm in previous literature [14]. After the optical identification of a monolayer, we transferred it on top of a hexagonal boron nitride (h-BN) flake previously exfoliated onto a SiO_2_ (285 nm)/Si substrate; the resulting stack is shown in Figure 1a. Then, we defined the device geometry by electron beam lithography (EBL), using a homemade PMMA resist (4% in chlorobenzene). After the EBL exposure, we developed the resist with a mixture of 1 part methyl isobutyl ketone (MIBK) to 3 parts isopropanol. Figure 1b shows the geometry of the device after the dry-etching process, in which we etched away the EBL exposed areas in a SF_6_ atmosphere using inductively coupled plasma reactive-ion etching (ICP-RIE). Once the geometry of the stack was defined, we performed a second EBL to define the contact geometry, and deposited Ti (10 nm)/Au (60 nm) contacts via e-beam evaporation in an ultrahigh vacuum (10^−8^ mbar). Figure 1c shows the device after evaporation of the metallic contacts. The final step consisted of transferring an h-BN layer on top of the device using a dry transfer method that relies on the use of polycarbonate film (PC) for stacking of van der Waals heterostructures [14,15]. Once the device was completed, we annealed the sample for 3 h at 200 °C in vacuum to eliminate bubbles trapped between the different layers; Figure 1d shows the resulting device. The encapsulation of the 1L-MoSe_2_ allowed us to obtain a high-quality phototransistor, improving the performance of the device, decreasing the influence of the impurities of the SiO_2_ substrate, and reducing the bandwidths of the spectral features [16].

### 2.2. Electrical Characterization of The Monolayer MoSe_2_ Phototransistor

We started our measurements by characterizing the electrical behavior of the device. Unless otherwise specified, all of the measurements described below were performed in vacuum and at *T* = 5 K. Figure 2a shows the gate transfer curve of the 1L-MoSe_2_ phototransistor at *V*_ds_ = 10 V, showing a clear n-type behavior with a threshold gate voltage at *V*_th_ = 13 V. Figure 2b shows the two-terminal *I–V* characteristic of the device at *V*_g_ = 33 V ($V_{g}-V_{\mathrm{th}}=20\;V$), exhibiting a nonlinear behavior due to the presence of asymmetric Schottky barriers in the contacts. Both measurements were almost hysteresis-free.

## 3. Results

### 3.1. Photocurrent Spectroscopy Measurements

Next, in order to characterize the 1L-MoSe_2_ spectral photoresponse, we exposed the whole device to a monochromatic light source of tunable wavelength [8]. Figure 3a schematically shows the low-temperature photocurrent setup used for the measurements. The sample was placed inside a pulse-tube cryostat with an optical access. The light source was a supercontinuum (white) laser (SuperK Compact, from NKT photonics), and the excitation wavelength was selected using a monochromator (Oriel MS257 with 1200 lines/mm diffraction grid); this enabled us to scan the visible and NIR spectral range, roughly from 450 nm to 1000 nm. The setup also included a halogen lamp and a CCD camera, aligned with the laser excitation via two beam splitters, allowing for easy sample alignment. In order to improve the signal-to-noise ratio of the optoelectronic measurements, an optical chopper modulated the excitation signal, and the electrical response of the device was registered using a lock-in amplifier with the same modulation frequency.

Figure 3b shows a typical photocurrent spectrum of the 1L-MoSe_2_ phototransistor, measured at $V_{\mathrm{ds}}=10\;V$ and $V_{g}=-40\;V$ ($V_{g}-V_{\mathrm{th}}=-53\;V$). The resulting spectrum can be fitted by a series of Lorentzian peaks, as depicted in the figure. We identified three main sharp spectral features at 1.616 ± 0.001 eV, 1.634 ± 0.001 eV, and 1.822 ± 0.004 eV, as well as two smaller features at 1.799 ± 0.010 eV and 1.705 ± 0.002 eV. The peaks at energies *E* = 1.634 ± 0.001 eV and *E* = 1.822 ± 0.004 eV correspond to the A and B excitonic ground states ($X_{1s}^{A}$ and $X_{1s}^{B}$) of 1L-MoSe_2_, respectively [17]. The peak at *E* = 1.616 ± 0.001 eV appears 19 meV below $X_{1s}^{A}$, as expected for the A trion transition $T_{A}$ [18]. Similarly, the feature at *E* = 1.799 ± 0.010 eV may correspond to the B trion transition T_B_, although it might also contain a contribution from the first excited Rydberg state of the A exciton, $X_{2s}^{A}$, which is expected to appear at a similar energy. The origin of the last feature, at 1.705 ± 0.002 eV, is not clear to us at this point, as it does not correspond to any of the typical features observed in low-temperature photoluminescence spectroscopy for 1L-MoSe_2_; however, we observed a similar exciton-like feature between the $X_{1s}^{A}$ and the $X_{1s}^{B}$ transitions in recently reported PCS measurements for h-BN-encapsulated 1L-MoS_2_ devices. This transition might correspond to a non-radiative interlayer exciton, forming at the interface between the 1L-MoSe_2_ and the h-BN. The uncertainty of the energies of the spectral peaks was extracted from the fitting errors.

As we will now discuss, the spectral features described above appear consistently in photocurrent spectra measured at different gate voltages.

### 3.2. Gate Dependence of The Photocurrent

Let us now investigate the gate voltage dependence of the photoresponse. In TMD-based phototransistors, photocurrent typically emerges from two different mechanisms: the photoconductive effect (PCE), where light-induced formation of electron–hole pairs leads to increased charge carrier density and electrical conductivity; and the photogating effect (PGE), where the light-induced filling or depletion of localized states causes a shift in the Fermi energy [19,20]. These two mechanisms can be distinguished by their different dependence on the gate voltage—while PCE-generated photocurrent typically depends weakly on *V*_g_, PGE produces a gate-voltage-dependent photocurrent proportional to the out-of-plane transconductance $G\equiv dI_{\mathrm{ds}}/dV_{g}$ [21,22,23].

Figure 4a shows the gate dependence of the photocurrent measured for illumination energy in resonance with the A exciton, *E* = 1.634 eV. For gate voltages above the threshold voltage (*V*_th,e_; see Figure 2a), we find that the measured photocurrent is proportional to the transconductance, as expected for PGE. However, for *V*_g_
*< V*_th_ a small but measurable photocurrent is still present, even when the transconductance drops to zero. The inset of Figure 4a shows a zoomed-in view of the region where the channel of the device is closed; there, the photocurrent remains in the range of 10 pA, and decreases monotonically as the gate voltage is lowered, until it cancels out at $V_{g}\approx -84\;V$. Remarkably, for gate voltages below −84 V, the photocurrent becomes negative, i.e., the device becomes more resistive upon illumination. As further discussed below, we associate $V_{g}\approx -84\;V$ (the voltage at which *I*_PC_ switches sign) with the charge neutrality point of the 1L-MoSe_2_ channel, $V_{0}$.

As we will now discuss, the exotic regime of negative photocurrent observed for $V_{g}<-84\;V$ can be well understood assuming that the photoresponse is dominated by PGE. Figure 4b schematically depicts a typical transfer gate ramp of an ambipolar TMD transistor in the dark (OFF) and under illumination (ON). When the Fermi level is in the band gap of the MoSe_2_, *I*_ds_ is zero due to the absence of free charge carriers. As the gate voltage increases (decreases) and reaches *V*_th,e_ (*V*_th,h_), *I*_ds_ shows an abrupt increase due to the filling of electrons (holes) in the conduction (valence) band. Upon illumination, photoexcited charge carriers may accumulate in localized states within the semiconductor bandgap, inducing a shift in the Fermi energy. This results in a horizontal shift of the transfer *I–V* curve by $\Delta V_{\mathrm{pge}}$. The light-induced photocurrent *I*_PC_ generated by this mechanism is given by
(1)$$I_{\mathrm{PC}}=\Delta V_{\mathrm{pge}}\frac{dI_{\mathrm{ds}}}{dV_{g}}\;$$

In Equation (1), $\Delta V_{\mathrm{pge}}$ may in principle have either a positive or negative sign, depending on the nature of the localized states involved in the PGE generation (either electron acceptor or electron donor). In monolayer MoSe_2_ phototransistors, $\Delta V_{\mathrm{pge}}$ is generally positive [24], as we also assume in the schematic drawing from Figure 4b. 

The transconductance $dI_{\mathrm{ds}}/dV_{g}$, on the other hand, changes sign as a function of *V*_g_, as it is positive for Fermi energies near the conduction band and negative near the conduction band. Thus, at negative gate voltages approaching the valence band threshold voltage, PGE is expected to yield a negative photocurrent, as we observed in our measurements.

### 3.3. Gate Dependence of Spectral Features

We now turn our attention to the evolution of the photocurrent spectra with $V_{g}$. In order to prevent inconsistencies due to photodoping of the device, which can result in a slow drift of the threshold voltage during the spectral acquisition [25], we measured the off-current of the device before every spectral ramp, and presented the gate-dependent spectra as a function of $V_{g}\;–\;V_{0}$ (further discussed in Appendix A). Figure 5a shows photocurrent spectra acquired at a fixed drain–source voltage ($V_{\mathrm{ds}}=10\;V$) for different gate voltages. As discussed above, $I_{\mathrm{PC}}$ flips sign as the gate voltage is modified. The four principal spectral features described above (see Figure 3b)—i.e., $T_{A}$, $X_{1s}^{A}$, $X_{1s}^{B}$, and $X_{2s}^{A}$—appear consistently in the different spectra; however, the position of these spectral features changes with *V*_g_. Gate-voltage-dependent changes in the position and intensity of excitonic features have also been reported in earlier literature for photoluminescence spectroscopy in monolayer TMDs such as MoS_2_ [6] and WSe_2_ [26]. 

Let us now focus on the evolution of the A exciton and trion transitions. Figure 5b shows the positions of the $T_{A}$ and $X_{1s}^{A}$ as functions of the gate voltage and the Fermi energy. There, we observe that the energy position of the $T_{A}$ transition changes abruptly at $V_{0}$, as the sign of $I_{\mathrm{PC}}$ inverts. A similar energy shift in the $T_{A}$ spectral feature has been reported in previous literature for MoSe_2_ [27,28], usually being attributed to the presence of two different trion species—either positively ($T_{A}^{+}$) or negatively charged ($T_{A}^{-}$)—whose relative spectral weights can be tuned by the gate voltages. More specifically, by changing $V_{g}$ one can modify the population of free charge carriers in the 1L-MoSe_2_ channel, favoring the interaction of photogenerated excitons with either free electrons or holes to form negative or positive trions, respectively. In our case, for $V_{g}<V_{0}$, where $T_{A}^{+}$ is expected to be the dominant trion species, the trion transition is centered at $E_{T_{A}^{+}}=1.620\pm 0.003$ eV, while for $V_{g}>V_{0}$, where $T_{A}^{-}$ should be dominant, we get $E_{T_{A}^{-}}=1.613\pm 0.005$ eV.

In addition to modulating the population of different trion species, increasing the gate voltage is also expected produce a monotonic blue-shift of the exciton energy (see Appendix A), while inducing a red-shift in the trion energy [29,30,31,32]. As we show in Figure 5b, this effect was also observed in our spectra.

Figure 5c shows the energy splitting between $X_{1s}^{A}$ and $T_{A}^{-}$ as a function of the gate voltage and the Fermi energy level. We found that the splitting between the two transitions increases linearly with the doping level of the device, as predicted and observed in previous literature for exciton and trion transitions [30,33,34], following the equation
(2)$$E_{X_{1s}^{A}}-E_{T_{A}^{-}}=E_{b}^{T_{A}^{-}}+cE_{F}$$
where $E_{X_{1s}^{A}}-E_{T_{A}^{-}}$ is the splitting between exciton and trion transitions; $E_{b}^{T_{A}^{-}}$ is the binding energy of $T_{A}^{-}$; $E_{F}$ is the Fermi energy, which is proportional to the back-gate voltage; and *c* is the slope of the linear fit, predicted to have a value of c≈1 [33]. 

To estimate the Fermi energy from the gate, we recur to a parallel-plate capacitor model, using the back-gate capacitance $C_{\mathrm{bg}}$ of the device, and considering a band effective mass of $m_{\mathrm{eff}}=0.8\;m_{0}\;$ [35], with $m_{0}$ being the electron mass (further discussed in Appendix A):(3)$$E_{F}=\frac{\hbar^{2}\pi \;}{2m_{\mathrm{eff}}e^{2}}\frac{C_{\mathrm{bg}}}{e}V_{g}\;$$

Using Equation (2) to fit the experimentally measured energy of the trion spectral feature, $E_{T_{A}^{-}}$ (see Figure 5c), we can estimate the binding energy of $T_{A}^{-}$, defined as the energy required to form a trion in the limit of infinitesimal doping [36]. To do so, we assume that the charge neutrality point corresponds to the gate voltage at which the photocurrent cancels out and the dominant trion species switches from $T_{A}^{+}$ to $T_{A}^{-}$ [29,37], i.e., $V_{0}$. Under this assumption we get $E_{b}^{T_{A}^{-}}=14.54\pm 0.79\;\mathrm{meV}$ and a slope c≈1.14±0.05. This value of the slope is consistent with earlier studies [30,38], which interpreted the pre-factor $cE_{F}\approx E_{F}$ as the additional energy required to place the charge from the dissociated trion on the top of the Fermi sea, as required by the Pauli blockade [33]. The values obtained here for $E_{b}^{T_{A}^{-}}$ and *c* are in good agreement with the large effective masses predicted by recent studies in Mo-based TMDs [8,17,35,39].

Recent literature on 1L-MoSe_2_ devices fabricated directly onto a SiO_2_ substrate gives slightly larger trion binding energies—around 20 meV [27]. Recently, $E_{b}^{T_{A}^{+}}$ was also measured by photoluminescence spectroscopy in h-BN-encapsulated MoSe_2_ grown by CVD [38]; there, the authors report an even larger trion binding energy, $E_{b}^{T_{A}^{+}}=27\;\mathrm{meV}$. It is also instructive also to compare the binding energy obtained for $T_{A}^{-}$ with the case of MoS_2_, where $E_{b}^{T_{A}^{-}}$ has been found to be 18 meV [30]—3.5 meV higher than the value obtained here for MoSe_2_; this energy difference is in good agreement with theoretical predictions [40].

The contrast between our results for $E_{b}^{T_{A}^{-}}$ and those given earlier literature may be related to the use of different substrates [16,41] and/or fabrication procedures. Furthermore, it is worth noting that, for gate-voltage-dependent measurements performed via fully optical spectral techniques, photodoping effects are not usually monitored, but may still be present, which could result in systematic errors when estimating the doping level.

## 4. Discussion

All in all, we studied the properties of the neutral excitons and different trions species in an h-BN-encapsulated MoSe_2_ phototransistor using low-temperature photocurrent spectroscopy, allowing us to obtain photocurrent spectra with excitonic linewidths of 15 meV, revealing that this technique is suitable and useful for the performance of spectroscopic analysis of TMDs. We fully resolved excitonic ground states ($X_{1s}^{A}$ and $X_{1s}^{B}$), trions related to $X_{1s}^{A}$ ($T_{A}$), and one excited state $(X_{2s}^{A}$). 

We explored the effects of doping on the excitonic transitions by measuring photocurrent spectra at different gate voltages, finding that the photocurrent switches signs as a function of the Fermi energy, being negative for gate voltages below $V_{g}\approx -84\;V$ and positive above this value. This change in sign is not frequently observed in TMD-based phototransistors, as gate voltages below –50 V are not commonly used for these devices, due to the risk of dielectric breakdown of the SiO_2_ insulating layer. However, a negative photoresponse is indeed expected assuming that PGE is the main generation mechanism for the photocurrent.

Finally, we also studied the effects of the gate voltage in the spectral position of the $X_{1s}^{A}$ and $T_{A}$ transitions. We observed an abrupt shift in the position of the T_A_ spectral features as the gate voltage crossed the neutrality point *V*_0_, which we attributed to a change in the dominant trion species, from negatively to positively charged trions ($T_{A}^{-}$ and $T_{A}^{+}$, respectively). For voltages above *V*_0_, we observed a continuous electrical tuning of the transition energies, induced by the change in charge carrier density in the material. By fitting the energy difference between the exciton and trion levels, we can estimate the trion binding energy for negative trions.

This work demonstrates low-temperature photocurrent spectroscopy as a powerful technique for the study of optoelectronics and exciton physics in two-dimensional systems. The results presented here provide new insight into the exciton-mediated optoelectronic response of 1L-MoSe_2_. 

## Figures and Tables

**Figure 1 nanomaterials-12-00322-f001:**
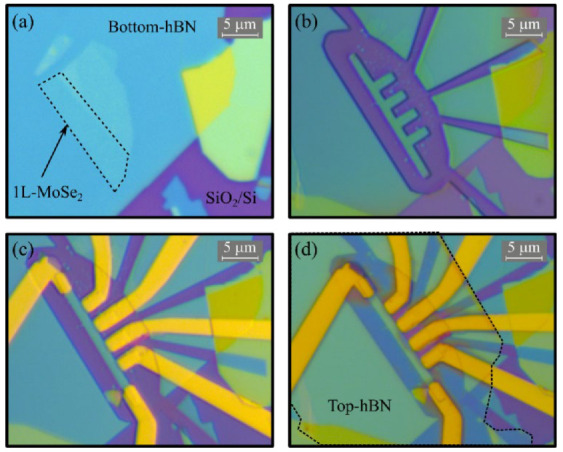
Optical microscope images of the 1L-MoSe_2_ device at different stages of the fabrication process: (**a**) Transfer of the 1L-MoSe_2_ above the bottom h-BN. (**b**) Definition of the device geometry after the dry etching. (**c**) Fabrication of the Ti/Au contacts. (**d**) Final device after the transfer of the top layer of h-BN.

**Figure 2 nanomaterials-12-00322-f002:**
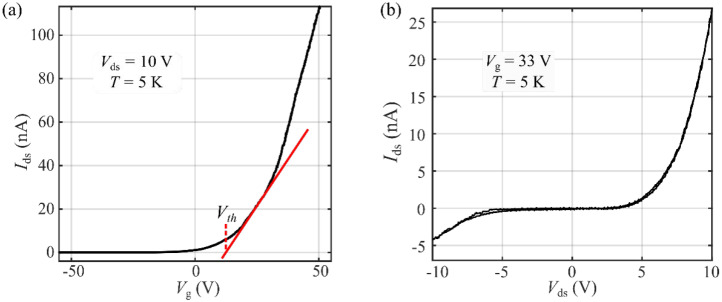
Electronic response and photocurrent spectrum of the device: (**a**) Gate transfer characteristics at *V*_ds_ = 10 V. The threshold gate voltage is estimated by extrapolating the linear region of *I*_ds_ (red solid line). (**b**) *I*–*V* characteristics of the device measured at *V*_g_ − *V*_th_ = 20 V.

**Figure 3 nanomaterials-12-00322-f003:**
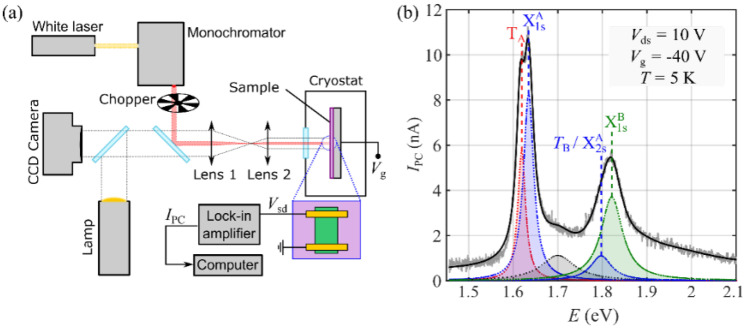
Photocurrent spectroscopy measurements: (**a**) Schematic of the low-temperature photocurrent setup. (**b**) Photocurrent spectrum of the MoSe_2_ phototransistor (grey line) and fitting to a multipeak Lorentzian function (black). The spectral features identified are depicted below the photocurrent spectrum.

**Figure 4 nanomaterials-12-00322-f004:**
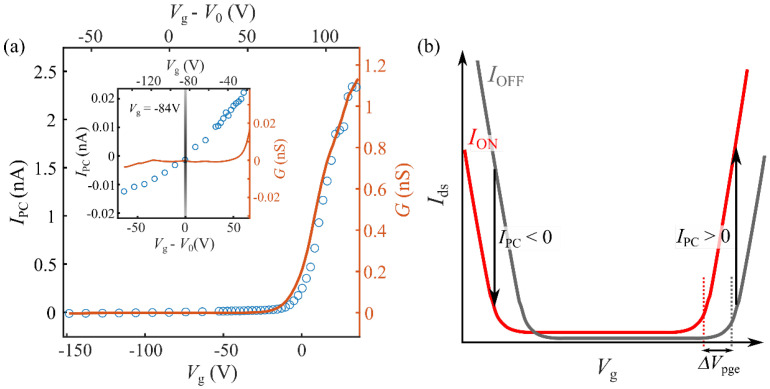
Gate dependence of the observed photocurrent: (**a**) Transconductance of the device (orange line, right axis) and gate-dependent photocurrent measured at *V*_ds_ = 10 V, in resonance with the exciton $X_{1s}^{A}$ transition, at a modulation frequency of f = 31.81 Hz. The inset shows a zoomed-in view of the region where the channel is closed. (**b**) Schematic gate ramps of an ambipolar transistor, under illumination and no illumination. The photoresponse contribution of Δ*V*_pge_ is depicted in the plot.

**Figure 5 nanomaterials-12-00322-f005:**
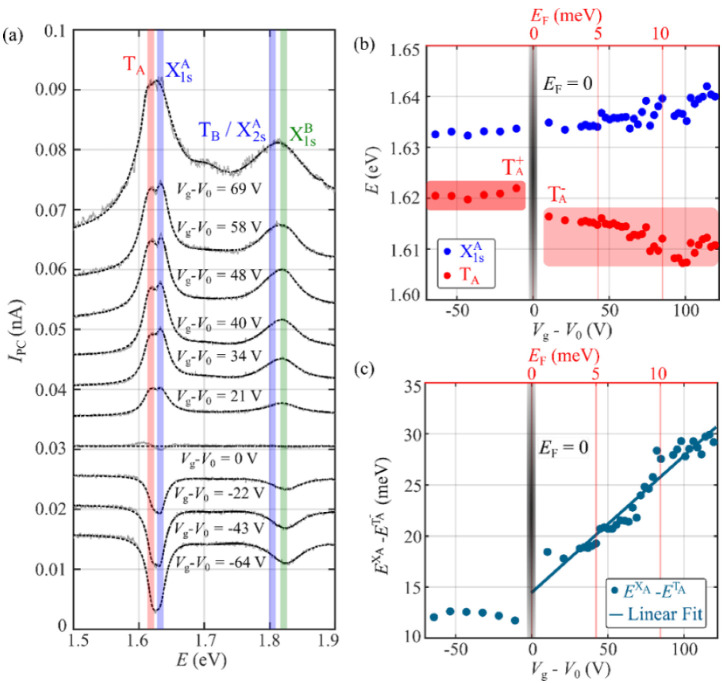
Gate dependence of the photocurrent spectra: (**a**) Individual photocurrent spectra (grey lines) acquired at different gate voltages, *V*_g_ − *V*_0_, in a range between −64 V and 69 V, fitted to a multipeak Lorentzian function (dashed black lines). For clarity, the spectra are vertically offset in steps of 0.005 nA. (**b**) Evolution with gate voltage of $X_{1s}^{A}$ and $T_{A}$ ($T_{A}^{+}{\;\mathrm{and}\;T}_{A}^{-}$) transition energies. (**c**) Gate voltage dependence of the energy splitting between the $X_{1s}^{A}$ and $T_{A}$ transitions.

## Data Availability

The presented data are available upon reasonable request to the authors.

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
