# Peer review of "The Low-Temperature Photocurrent Spectrum of Monolayer MoSe2: Excitonic Features and Gate Voltage Dependence"

_nanomaterials, 2022, doi:10.3390/nano12030322_

Round 1
Reviewer 1 Report
Vaquero et al. are reporting on the importance of using low temperature photocurrent spectroscopy for studying excitons in low dimensional materials. Specifically metal dichalcogenides of MoSe2 are investigated. The MS is concise discussing some fundamental and important aspects related to photocurrent and photogating effects using felid effect transistors and at low temperature. It is of a high quality and does not require any major experimental changes or concerns but mostly organization in the text. Please find below some minor points to consider.
- Page 4, line 107-~111, I would highly recommend considering errors, and therefore, error values, to be added to the provided photocurrent measurements, such as, 1.616,… . maybe a peak fitting error can be good to be used.
- Line 140, “until it cancels out at ? ≈ −84 V” this is not clear in the figure.
- Page 4, line 129-132, “These two mechanisms can be distinguished by their different dependence on the gate voltage: While PCE-generated photocurrent typically depends weakly on Vg, PGE…” Please add references.
- Materials and methods, section 2.3, starting from line 107, is providing results and discussion. I would highly recommend this to be moved to later sections as it is not materials or methods.
- Section 4 seems to be a long conclusion or summary. I would recommend to add a simpler and concise conclusion or summary. As it is now, it seems that there is no conclusion or summary.
- Please add description or introduce the used abbreviation “ICP-RIE” “Inductively Coupled Plasma - Reactive Ion Etching”.
- Similarly, please introduce “h-BN” “hexagonal-BN” and I would recommend to add the reason why it was used.
- Please note that figure 3 has (c) and (d) not (a) and (b) please correct it.
Reviewer 2 Report
The authors have done a wonderful job with the manuscript about low temperature photocurrent spectrum study of monolayer MoSe2. The paper is very well written with sufficient experimental results. I have here few comments to add before publication.
1. Please write process of deposition of MoSe2 on PDMS substrate in brief? Full form of PDMS may be useful.
2. What is the thickness of mono layer MoSe2? Did you try MoSe2 of different thicknesses to understand the behavior of the material? Does thickness of the material matter for your study?
3. Did you use metal mask to deposit Ti/Au layer? What was the thickness of Ti and Au layers deposited on top of MoSe2? What is the minimum thickness of Ti necessary for uniform results?
4. Is MoSe2 a p-type semiconductor? What are band gap and work function of this material? What types of junction is made between MoSe2/Ti/Au? It looks like work function of MoSe2 and Ti are quite similar and you expect ohmic contact. Is this the reason you chose Ti before deposition of Au?
5. What is the role of Au deposition? If instead of Ti, Au was deposited directly on top of MoSe2, would you get the same results?
6. You have performed the experiment at low temperature. Please add few sentences about the significance of using low temperature measurements? What would happen if this study was made at room temperature?
7. The authors are requested to add conclusion section in the manuscript where they can explain importance of this paper for scientific community. You have explained to some extent about the need of this work in introduction section. Now, in conclusion section, please add few sentences how your work compensated what was missing before this manuscript.
